# Involvement Theory with Market Segmentation: Effect of False Functional Food Advertising on Purchase Intention

**DOI:** 10.3390/foods11070978

**Published:** 2022-03-28

**Authors:** I-Hsuan Wu, Chaoyun Liang, Ching Yin Ip

**Affiliations:** 1Department of Bio-Industry Communication and Development, National Taiwan University, Taipei 10617, Taiwan; d08630003@ntu.edu.tw; 2Division of Quality Compliance and Management, Food and Drug Administration, Ministry of Health and Welfare, Taipei 10617, Taiwan; 3Department of Advertising and Public Relations, Fu Jen Catholic University, New Taipei City 242062, Taiwan; ipchingyin@yahoo.com.hk

**Keywords:** advertising involvement, functional food, health-related claims, product involvement, purchase intention, situational involvement

## Abstract

In certain cases, people’s health can be compromised or medical treatment delayed as a result of their misplaced belief in false advertisements and purchasing of functional foods. These advertisements can be divided into three distinct types of claims: nutrition, health, and reduction in disease risk. This study analysed how, after consumers realise advertising violations, their intention to purchase functional foods with different claims are affected by the degree of consumer involvement (product, advertising, and situational involvement) and region of residence. A total of 1046 survey responses were collected for analysis. The results reveal that both product and advertising involvement influence purchase intention through the mediation of situational involvement. Residents in nonnorthern regions of Taiwan exhibited a greater effect of overall involvement on purchase intention than did those in the north. In addition, products and advertisements with health claims had a stronger effect on purchase intention than did those with nutrition and disease risk reduction claims. The results indicate that, for functional foods and advertisements with nutrition and health claims, the effects of overall involvement on nonnorthern residents’ purchase intentions were greater than those on the northern residents, but for functional foods with disease risk reduction claims, the effects were greater on the northern residents’ purchase intentions.

## 1. Introduction

Food has long been used to prevent and treat disease in Asian countries where people perceive particular foods as traditional [1]. In addition to providing nutrition, functional foods can improve health and reduce disease risks [2]. In the last two decades, noncommunicable chronic diseases have been prevalent globally, causing severe disease burden, medical trauma, and increased expenditure [3]. Improvement in public health has become an obligatory policy for governments of various countries, with numerous plans and guidelines related to the development of functional foods issued to promote innovation and encourage the food industry to provide people with healthier choices [4].

Functional food claims (i.e., nutrition, health, and disease risk reduction) are crucial to consumers who measure the health benefits of certain products and make purchase decisions based on these claims [4]. Research has indicated that functional foods with health-related claims are considered healthier than products without such claims [5], which emphasises the key role of health-related claims in functional foods. Because public awareness of health promotion through diet has increased, consumers are more inclined to purchase functional foods at a premium compared with traditional foods [1]. As the consumer market for functional foods expands, the functions and advertisement of these products have become major topics [6].

Food advertising affects consumer preferences, purchase intention, and eating behaviour [7]. The advertisements for functional food often employ the credibility, influence, and persuasive power of celebrities and authorities to increase consumers’ purchase intention [8]. However, false or misleading advertisements can result in inappropriate or unnecessary participation in health services and negatively affect people’s self-care decisions. The consumption of foods that claim to reduce the risk of disease may be used as a replacement for formal medical care [9]. People who lack medical resources often engage in self-care behaviours [10]. Chang and Tung indicated that the decision to consume functional food is associated with customers’ residential region, with urban women exhibiting more willingness to purchase functional food [11]. The positive association between functional food purchase and customer residential region is further supported by several studies [12,13,14]. Therefore, in discussions of topics related to purchasing functional foods, regional differences must be taken into account.

With changing social patterns, the improving quality of life, and advent of an aging society, health has become a dominant topic among Taiwanese people, driving the market demand for functional foods. In 2017, Taiwan’s functional food market was valued at approximately USD 4.5 billion, with a growth rate of 6.43%; in 2018, the growth rate was 5.88% [15], whereas the global market size was estimated to be USD 177.770 million in 2019 and is expected to reach USD 267,924.4 million by 2027 [16]. Maintaining or losing weight is often a particular concern of Taiwanese women, and annual spending on functional foods for weight control has reached USD 35 million in the past 3 years. However, in recent years, Taiwan’s health authorities have reported over 4500 food advertisement violations annually, for which they have issued fines as high as USD 4 million. Such advertisements often claim false effects such as weight loss or anticancer properties, which can lead to consumers losing confidence in functional foods, weakening their purchase intention. In Taiwan, regional differences are evident between ‘small areas of land with abundant [medical] resources’ and ‘large areas of land with scarce [medical] resources’ [17]. A large gap in terms of medical resources is present between the northern and nonnorthern regions of Taiwan [18].

A high degree of consumer involvement can generate strong purchase intention and predict consumer purchase behaviours, which is reflected in numerous countries including Taiwan [19,20]. Consumer involvement can be divided into product, advertising, and situational involvement [21]. Product involvement represents an experience with symbolic value in which different products incorporate varying degrees of involvement that are the precursors of purchase decisions [22]. Advertising involvement, referring to the personal state induced by specific stimuli, directly affects the effectiveness of advertising [23,24]. Situational involvement is related to short-term changes, such as health status, in a consumer’s environment that largely affect purchasing behaviour [25]. Involvement is a robust predictor of purchasing decision [26], but few studies have provided integrative analyses on different involvement types, which warrants further inquiry.

To narrow the research gap and obtain a deeper understanding of the effect of false advertisements on consumers’ intention to purchase functional food, the influence of different consumer involvement types on purchase intention must be clarified; the interaction of involvement types and market segmentation must also be assessed. This study also evaluated consumers’ responses after advertisements were revealed to be false. The false advertisements used were adopted from the penalty list issued regularly by the Taiwanese government. The following three research questions were addressed: (i) What relationship can be observed among the three different involvement types? (ii) What different effects do product, advertising, and situational involvement have on purchase intention? (iii) Are the health-related claims (i.e., those concerning nutrition, health, and disease risk reduction) in false advertising and residential region associated with purchase intention? The research outcomes act as a reference for government agencies for planning and enacting appropriate management measures to protect consumer rights and promote public health.

## 2. Literature Review

### 2.1. Involvement Theory and Possible Construct

Involvement theory holds that individuals direct their attention to stimuli or circumstances with which they perceive themselves to be related [26]. Involvement, particularly consumer involvement in health, is a construct of human motivation related to personal values and needs [26]. Involvement, particularly consumer involvement in health, is a construction of human motivation related to personal values and needs [26]. A high degree of consumer involvement generates stronger purchase intention; that is, self-involvement in purchasing can predict a consumer’s buying attitudes and behaviours [19]. Involvement can be classified into product, advertising, and situational involvement [21]. Product involvement refers to the extent to which an individual attaches meaning to a product or service [22]. Advertising involvement refers to individuals’ perceived connection to the message of an advertisement and their psychological state when exposed to an advertisement [23,24]. Situational involvement refers to a short-term perceived connection to a product or service based on a consumer’s personal needs [25].

Certain key features (such as health-related claims) can increase product involvement considerably [27]. Health-based motivations positively affect consumer involvement, and highly involved shoppers are likely to enjoy searching for products and deals [28,29]. Scholars have indicated a need to customise communication and promote functional foods through consumer involvement by considering the health motivations behind consumers’ purchase decisions [30]. When consumers become more involved in a particular advertisement, their awareness of the health benefits of functional foods increases along with their purchase intention [3]. Furthermore, studies have demonstrated that situational involvement is associated with external stimulus preferences. When individuals or their friends experience health problems, their perception of health risks and motivation for disease prevention subsequently increase. Therefore, situational involvement has a positive effect on the intention to purchase functional foods [31].

Many people search the Internet for relevant information prior to purchasing. Visual attraction and derived enjoyment are essential elements of advertising involvement [32,33] that positively affect situational involvement. Scholars have also noted that product involvement can affect brand commitment through the mediator of situational involvement [25,34]. Although purchase involvement is essentially equivalent to situational involvement, the scope is limited to the context of the purchase process. Therefore, we proposed the following hypotheses.

**H1.** 
*Situational involvement is positively associated with intention to purchase functional food.*


**H2.** 
*Product involvement is positively associated with intention to purchase functional food through the mediator of situational involvement.*


**H3.** 
*Advertising involvement is positively associated with intention to purchase functional food through the mediator of situational involvement.*


### 2.2. Current Market for Functional Foods and Consumption Intention

Demand for functional foods is rising rapidly, and the market has great potential for development [35]. Functional foods generally refer to foods with one or more particular functions and positive effects on human health, often in relation to nutrition or reducing disease risk [36]. The growth of the functional food market has been largely caused by concerns about rising medical and healthcare costs, the desire to improve the quality of later life with the steady increase in life expectancy, and scientific evidence that functional foods are beneficial to health [37]. Purchase intention, which is strongly associated with market expansion, is the likelihood that consumers plan to purchase specific products or services. Consumer attitude towards a certain product or brand, along with certain external factors, promote consumers’ purchase intentions and are crucial indicators for predicting consumer behaviour [38]. Research has revealed that health-related claims in functional food advertisements, market segments, and government policies and laws all affect purchase intention of these foods [39].

According to the Codex Alimentarius Commission [40], the health-related claims of functional foods can be divided into those of nutrition, health, and reduction in disease risk. Nutrition claims refer to specific stated or implied nutritional properties, and health claims state or imply a specific relationship between certain foods and an aspect of health; claims to reduce disease risk refer to statements that the consumption of certain foods is associated with a reduced risk of developing or contracting diseases [40]. Nutritional information and health-related claims have positive effects on purchase intention, even when products are sold at a premium [41]. Food categories and health-related claims, separately and in combination, are essential factors affecting consumer perception and acceptance of functional foods. A product category regarded as healthier generally has a high degree of consumer acceptance, particularly for products with disease risk reduction claims [3].

Regional differences are crucial factors affecting self-care practices, particularly in relation to the allocation of medical resources [42]. The uneven development and supply of medical services between urban and rural areas have resulted in a low utilisation rate of medical services in rural areas and a high proportion of inhabitants exercising self-care behaviours [10,43]. Studies have reported that the lack of high-quality and dependable medical services has often driven disadvantaged groups towards convenient, medically unconventional treatments, including the intake of functional foods as a substitute for medicinal drugs and professional medical advice, that can induce adverse health effects [42]. Especially for people living in remote areas that lack modern medical facilities and advanced treatments, functional foods are regarded as having disease prevention and treatment properties as well as nutritional benefits [44]. However, research on functional food purchasing in relation to demographic variables has mostly centred on gender, age, education level, living situation, and marital status [45,46] but has rarely discussed regional differences. Therefore, we proposed the following hypotheses and the conceptual model (Figure 1):

**H4.** 
*A significant difference in functional food purchase intention is present between regions in Taiwan as a result of medical resource availability.*


**H5.** 
*A significant difference in functional food purchase intention is present among the three distinct claim types.*


## 3. Methods

This study investigated the effects of different consumer involvement types on purchase intention. The mediating and moderating effects of situational involvement and market segmentation (i.e., residential regions and health-related claims), respectively, were tested according to the hypotheses. To distinguish different health-related claims, we developed three separate questionnaires with similar question items. General Taiwanese consumers were targeted in this study, with 1046 valid samples collected through 3 online surveys; the samples comprised 350, 350, and 346 nutrition, health, and disease reduction claims, respectively.

The questionnaires contained four scales, one each for product involvement, advertising involvement, situational involvement, and purchase intention, as well as sections on false advertisement and demographic variables. More details of each construct are presented as follows:Product involvement scale: Based on Roe and Bruwer [22], Te’eni-Hararis and Lehman-Wilzig [47], and Zaichkowsky [48], this scale consisted of eight items. Representative items of nutrition claims include ‘Foods with nutrition claims are important to me’, ‘Foods with nutrition claims are valuable to me’, and ‘Foods with nutrition claims are attractive to me’.Advertising involvement scale: In reference to the literature [21,23,24,48], this scale had a total of six items. Representative items of nutrition claims include ‘Advertisements claiming certain foods are nutritious catch my attention’, ‘Advertisements claiming certain foods are nutritious appeal to me’, and ‘To understand the value of food, I spend time researching the nutritional benefits stated in the advertisements’.Situational involvement scale: In accordance with the suggestions of other studies [25,49], this scale comprised six items. Representative items of nutrition claims include ‘When purchasing foods claiming to have nutritional benefits, I believe I am making the right choice’, ‘I enjoy the process of purchasing foods claiming to have nutritional benefits’, and ‘Spending time shopping for foods claiming to have nutritional benefits is worthwhile’.False advertisements and penalties: A false advertisement with a particular health-related claim and the issued penalty was displayed directly after the situational involvement scale. The participants were asked to read the advertisement and penalty before completing the subsequent purchase intention scale. The false advertisement with nutrition claims focused on a high-protein, low-fat product; that with health-related claims centred on weight loss and weight loss function; and that with disease risk reduction claims focused on anticancer effects. The false advertisements used were drawn from the penalty list issued regularly by the Taiwanese government.Purchase intention scale: Based on the suggestions of Spears and Singh [50] and Te’eni-Hararis and Lehman-Wilzig [47], this scale comprised four items. Representative items of nutrition claims include ‘I will consider purchasing food with nutrition claims’, ‘I will continue purchasing food with nutrition claims’, and ‘I will recommend that my friends buy food with nutrition claims’.Demographic variables: This part contained basic information on gender, age, education level, and residential regions.

Purposive sampling was adopted to select the respondents of an online questionnaire survey. To ensure sampling adequacy, the study respondents were clearly informed of the research objective and their rights on the front page of each online questionnaire. Furthermore, this information stated that the collected data would be anonymised for subsequent analysis to assure respondents of their privacy. The information also stated clearly that participation was voluntary. The participants consented to participate and had the right to revoke that consent at any time. This declaration encouraged participants to provide accurate responses.

Our evaluation method was adapted from that of another quantitative study [51]; the questionnaire was composed of single-response questions rated on a 6-point scale (1 = strongly disagree; …; 6 = strongly agree). The survey was designed to prevent respondents from advancing to the next question if any information was missing; the returned questionnaires thus included no missing values.

Data collection was conducted using online questionnaires uploaded to the SurveyCake platform (see Appendix A). After a survey is uploaded, SurveyCake generates a uniform resource locator for the survey that can be distributed to target groups (e.g., fora on health promotion, functional foods, nutrition, and disease risk) to encourage participation and satisfy the target criteria. The online survey was conducted between 10 January and 10 March 2021. We removed questionnaires with zero or slight variance and those indicating straightlining or exhibiting contradictory responses. We analysed the questionnaire data by using SPSS for Windows 21.00 to describe the demographic variables, followed by confirmatory factor analysis and structural equation modelling (SEM).

## 4. Analyses

### 4.1. Descriptive Statistics

The demographics of the valid respondents in this study are described in Table 1. The survey sample was compared with the population data of Taiwan (approximate total population of 23.56 million, with 21.60 million aged above 15 years) to evaluate its representativeness based on a 95% confidence level and 95% reliability. The disparity between our sample and the national census can be attributed to the following factors: (1) women have a higher probability of buying functional foods than men [13,52,53]; (2) people aged between 20 and 40 years consume more functional foods than other age groups [52,54,55]; (3) consumers with a higher education level tend to consume more foods with health benefits than those without such education [54,56,57]; and (4) geographical region is a predictor of functional food purchase [12,13,14]. These factors assist in explaining why the survey response rates of certain groups were higher than those of other groups.

### 4.2. Confirmatory Factor Analysis

Confirmatory factor analysis was performed using SPSS 21 and Amos 21 to evaluate the factor structures of the scales based on the criteria suggested by Hu and Bentler [58]. The results demonstrate a satisfactory fit, with the following indices: χ^2^/*df* = 5.98, root mean square error of approximation (RMSEA) = 0.07, standardised root mean square residual (SRMR) = 0.04, comparative fit index (CFI) = 0.95, and Tucker–Lewis Index (TLI) = 0.94. The factor loadings of each item were between 0.61 and 0.97, indicating that each construct had satisfactory convergent validity. The composite reliability of each construct was between 0.64 and 0.80, which indicated strong reliability (Table 2). The heterotrait–monotrait (HTMT) ratio of the correlation results are reported in Table 3. The recommended HTMT threshold of 0.9 was used to provide sufficient evidence of discriminant validity [59,60].

Another common assessment of discriminant validity is the Fornell–Larcker criterion. According to Franke and Sarstedt [61], the use of average variance extracted (AVE) in the Fornell–Larcker criterion focuses on item loadings but ignores overall composite reliability, which is influenced by the number of indicators and their AVE. The HTMT is a more comprehensive and less constrained approach for assessing discriminant validity in SEM research than other techniques and is universally applicable to all latent variable methods [59]; hence, discriminant validity in this study was verified.

### 4.3. SEM

The maximum likelihood estimation of SEM models was employed in this study and the results demonstrate that the mediating model fit well with the collected data (χ^2^/*df* = 5.94, RMSEA = 0.07, SRMR = 0.04, CFI = 0.95, and TLI = 0.94). Mediation effects existed because the beta coefficient of the indirect effect of product involvement or advertisement involvement on purchase intention through situational involvement is significant [62]. Samples have multivariate normality if the Mardia coefficient is less than *p*(*p* + 2), where *p* is the number of observable variables. The Mardia coefficient for the hypothesised structural equation model was 268.55, which is less than 624. Therefore, maximum likelihood estimation could be applied to the SEM analyses.

## 5. Results

### 5.1. Mediating Model for H1, H2, and H3

The SEM results (Figure 2) supported H1, H2, H3, and the proposed mediation model. The standardised path coefficient of the path from product involvement to situational involvement was 0.46 (Cohen’s f^2^ effect size = 0.06), and that of advertising involvement to situational involvement was 0.44 (f^2^ = 0.12); product involvement and advertising involvement had a significant direct effect on situational involvement (*R*^2^ = 0.76). Furthermore, the standardised path coefficient of the relationship between situational involvement and purchase intention was 0.33 (f^2^ = 0.03), and situational involvement had a significant direct effect on purchase intention (*R*^2^ = 0.19). The standardised paths between product involvement and purchase intention and between advertising involvement and purchase intention were nonsignificant.

### 5.2. Mediating Model for H4

The results indicate no significant differences resulting from gender, age, or education level, though a significant difference was observed for residential regions. To test H4, residential regions were classified into northern and nonnorthern groups and then examined using a *t* test and invariance test. The *t* test results indicate that the purchase intention of the nonnorthern group was significantly greater than that of the northern group (*t* = 3.456, *p* value = 0.001). In addition, configural invariance was achieved because the unconstrained model for the two groups yielded an appropriate fit (χ^2^/*df* = 3.72, RMSEA = 0.05, SRMR = 0.04, CFI = 0.94, and TLI = 0.94). Metric invariance was further examined through a comparison of the chi-squared test results between the unconstrained and constrained (equal factor loadings) models; the results reveal a nonsignificant difference (Δχ^2^ = 30.15, Δ*df* = 20, *p* > 0.05) between northern and nonnorthern groups. Therefore, full invariance was observed.

The moderating role of residential region in the relationship between situational involvement and purchase intention was assessed through multigroup analysis of the northern and nonnorthern groups and application of the χ^2^ difference test in the constrained and unconstrained models. Moderation existed if the constrained model exhibited a significant positive change in χ^2^ compared with the unconstrained model [63]. The results reveal a significant change in χ^2^ between the constrained and unconstrained models (Δχ^2^ = 8.31, Δ*df* = 1, *p* < 0.004), indicating the presence of the moderating effect. The effect of situational involvement on purchase intention in the nonnorthern group (*β* = 0.25, *p* < 0.001, f^2^ = 0.01, Figure 3a) was weaker than that in the northern group (*β* = 0.34, *p* < 0.001, f^2^ = 0.03, Figure 3b). The SEM results illustrated in Figure 3 support H4 and the proposed moderation model.

### 5.3. Mediating Model for H5

To test H5, three health-related claims were examined through analysis of variance (ANOVA) and an invariance test. The ANOVA results indicate that the purchase intention associated with both nutrition and health claims was significantly greater than that associated with the disease reduction claim (*F* = 18.389, *p* value = 0.000, Scheffé test = nutrition and health > disease risk reduction). Moreover, configural invariance was achieved because the unconstrained model for the two groups yielded an appropriate fit (χ^2^/*df* = 3.00, RMSEA = 0.04, SRMR = 0.05, CFI = 0.94, and TLI = 0.93). Metric invariance was further examined through a comparison of the chi-squared test results between the unconstrained and constrained (equal factor loadings) structural models; the results reveal a nonsignificant difference (Δχ^2^ = 59.96, Δ*df* = 40, *p* < 0.05). Without the constraint for the three items in the scale, this difference became nonsignificant (Δχ^2^ = 47.78, Δ*df* = 34, *p* > 0.05); hence, partial invariance was observed.

The moderating role of advertising claims in the relationship between situational involvement and purchase intention was also tested through a multigroup analysis of three groups (nutrition, health, and disease risk reduction claims). A significant change in χ^2^ was revealed between the constrained and unconstrained models (Δχ^2^ = 3.48, Δ*df* = 2, *p* > 0.05) as a result of the moderating effect. The effect of situational involvement on purchase intention in the nutrition group (*β* = 0.45, *p* > 0.001, f^2^ = 0.05, Figure 4a) was stronger than that in the health group (*β* = 0.20, *p* < 0.001, f^2^ = 0.01, Figure 4b) and disease risk reduction group (*β* = 0.37, *p* > 0.001, f^2^ = 0.03, Figure 4c). The SEM results illustrated in Figure 4 supported H5 and the proposed moderation model.

To assess the effect of different residential regions for each advertising claim, residential regions were divided into the northern and nonnorthern groups. The results of the configural invariance and metric invariance tests are detailed in Table 4. The moderating effects of residential regions in the relationship between situational involvement and purchase intention were assessed through multigroup analysis. The analysis results are also presented in Table 4.

## 6. Discussion

In contemporary society, where chronic diseases are prevalent, life expectancy is increasing, and medical care costs are high, people pay more attention to functional foods because they not only meet consumer needs but are recognised as valuable. Consumer appreciation of functional foods is based on their positive expectations of the product’s unique healthcare functions to support them in times of illness and in later life. When people perceive that a particular food is essential, they become involved. Through scientific verification and media promotion, an increasing number of people believe that the ‘right’ food can benefit their health; therefore, demand for functional foods has risen sharply. Health-related claims have become an effective means of communicating the benefits of certain foods, with consumers believing that their subsequent purchase decisions are ‘informed’. With these factors taken into account, the following sections discuss the hypotheses tested in this study.

### 6.1. Effects of Involvement on Purchase Intention

Purchasing behaviour involves a series of activities spontaneously initiated by consumers, such as information retrieval, information browsing, product selection, and order placement. When consumers purchase functional foods, health promotion is likely the main motivation. By immersing themselves in the promising outcomes of the product, consumers perceive the benefit of health improvement during the purchase process. The results indicate that situational involvement is positively associated with purchase intention and that product involvement is positively associated with both purchase intention and situational involvement. Although consumers with a high degree of product involvement would expend more effort in retrieving and evaluating product information and tend to make rational decisions about product purchases, a study discovered that consumer perceptions of products and brands are often inferior to emotional marketing [64]. Consumers wish to obtain the expected benefits, such as health functions, from purchasing a product, and their awareness of the risk of false purchases therefore increases. They tend to repeatedly inquire about products during the purchasing process, increasing their degree of situational involvement [65]. With this high degree of situational involvement coupled with emotional marketing in stores, consumers consequently increase their purchase intention [66]. Compared with other research, this study clarified the relationship between product involvement and purchase intention and identified the mediating role of situational involvement.

The results indicate that advertising involvement is positively associated with both purchase intention and situational involvement; thus, advertising involvement may affect purchase intention through situational involvement. A high degree of advertising involvement reflects consumer engagement with the substantive content of an advertisement to which they devote more attention. Conversely, a low degree of advertising involvement reflects nonsubstantive content (e.g., when the professionalism and attractiveness of the information source are more influential) [49], indicating that advertising involvement is associated with information processing and application. Though advertising involvement has an inducing effect on purchasing behaviour, an active mediator is required to influence purchase intention [67]. The higher the degree of advertising involvement, the stronger the persuasiveness is of positive messages (such as health or disease risk reduction claims) to consumers [68]. This causes consumers to pay more attention to functional foods in specific situations. When consumers are highly involved in and enjoy purchasing, purchase intention is reinforced. Compared with other studies, this research further determined that situational involvement could generate a mediating effect between advertising involvement and purchase intention.

### 6.2. Effect of Residential Regions and Health-Related Claims on Purchase Intention

The results demonstrate that consumers living in different regions have differing degrees of situational involvement on purchase intention, with a higher degree observed in nonnorthern regions than in northern regions of Taiwan. Functional food has become a supplementary or alternative medical programme for contemporary people, and this mode of self-care has attracted much attention [42]. Medical resources and personnel are concentrated in Northern Taiwan, representing an unbalanced regional distribution [18]. Differences in regional medical resources are a critical factor affecting self-care practices [10,42,43]. Therefore, disadvantaged groups (i.e., nonnorthern residents) who lack medical resources often adopt unconventional self-care strategies (such as purchasing and eating functional foods). These results indicate that the effect of advertising involvement on the situational involvement of nonnorthern residents was stronger than that of northern residents. Because of the lack of medical resources in nonnorthern regions, health information is less likely to be communicated through formal medical channels, with advertisements serving as the main source of relevant information [6]. The results also reveal that the effect of product involvement on the situational involvement of northern residents was greater than that of nonnorthern residents. The political and economic centre of Taiwan is located in the north, where medical information is readily available. The people in this area are generally of high social and economic status and can acquire information from other regions [69]; thus, product and situational involvement in relation to functional foods are relatively high.

Consumers’ trust in different health-related claims varies [3] and generates different perceived benefits in the purchasing context. The effect of situational involvement on purchase intention also varies on the basis of health-related claim, which is consistent with the results of Verbeke, Scholderer, and Lähteenmäki [70]. For functional foods with nutrition claims, the effect of advertising involvement on situational involvement is greater than that of product involvement on situational involvement. Because functional foods with nutrition claims only claim or imply that they have specific nutritional properties, but fail to verify such direct health effects, consumers must rely on advertising messages that communicate their health benefits [6]. This study provides empirical evidence from Taiwan that the effect of situational involvement on the intention to purchase functional foods is significant when such foods have nutritional claims and that the effect of advertising involvement is stronger than that of product involvement. The results also reveal that, for functional foods with health claims, the effect of product involvement on situational involvement is stronger than that of advertising involvement on situational involvement. Functional foods with health claims suggest a particular relationship between certain foods and an aspect of health, and consumers acquainted with this relationship may be willing to pay a premium [41]. This study provides empirical evidence from Taiwan that functional foods with health claims also have a significant effect of situational involvement on purchase intention and that the effect of product involvement is stronger than that of advertising involvement. For functional foods with disease risk reduction claims, the effect of product involvement on situational involvement is slightly greater than that of advertising involvement on situational involvement. To achieve disease prevention and treatment, most people consult medical professionals or carefully read product information rather than relying solely on advertisements as the basis for their evaluation [71].

After residential regions and health-related claims were incorporated into the analysis, the results demonstrated that, in regard to functional foods and advertisements with nutrition and health claims, the effect of overall involvement on the purchase intention of nonnorthern residents was greater than that of northern residents. This phenomenon can be explained by the differences in regional medical resources and the degree to which residents rely on advertisements as the main source of health information [6]. For functional foods and advertisements with claims of disease risk reduction, the effect of overall involvement on the purchase intention of northern residents was greater than that of nonnorthern residents. Northern residents are generally of high socio-economic status and enjoy abundant medical resources; they are more likely to pursue health benefits, pay a premium for products, and bear the risk of mispurchase with more ease than nonnorthern residents [69]. In addition, the advertising involvement of northern residents is slightly higher than product involvement, which may be attributable to their ability to bear the risk of mispurchase, which reduces perceptions of potential risk and the effect of product involvement on situational involvement and purchase intention. In anticipation of obtaining more efficacious products, consumers’ engagement with the disease reduction risk claims in advertisements further promotes their situational involvement and purchase intention, which may also explain the strong influence of advertising involvement.

## 7. Conclusions and Suggestions

This study analysed the association of different involvement types on the intention to purchase functional foods with different health-related claims and accounted for the regions in which consumers resided. The results reveal that both product and advertising involvement are positively associated with situational involvement, and consequently, situational involvement is positively associated with purchase intention; that is, both product and advertising involvement are positively associated with purchase intention through the mediation of situational involvement. In addition, the effect of overall involvement on nonnorthern residents’ purchase intentions was greater than that on northern residents as a result of their differing medical resources. Products and advertisements with nutrition claims had a greater influence on purchase intention than those with health and disease risk reduction claims. The results demonstrate that the effect of overall involvement on nonnorthern residents’ purchase intentions was greater than that on northern residents in terms of functional foods and advertisements with nutrition and health claims; however, the effect of overall involvement on the northern residents’ purchase intentions was greater than that on the nonnorthern residents for functional foods with disease risk reduction claims.

The management of functional foods is increasingly crucial for public health. Mild advertising violations result in reduced consumer expectations towards a product and economic losses, but more serious violations lead to delays in medical treatment and potential health damage, which indicate the necessity of this study. Other research has focused on the effect of a particular involvement on purchase intention and the difference in consumer perception and acceptance of functional foods among diverse demographic populations and various health-related claims. This study analysed the complex interactions among different involvement types in depth, and the results not only revise the structural relationship of product, advertising, and situational involvement but also revealed the mediating role of situational involvement on purchase intention. In addition, this study distinguished the resultant effects of different advertising claims of functional foods, incorporating an analysis of consumers’ regions of residence and discussion of the national distribution of medical resources. Overall, we established a strong theoretical framework for consumption behaviour regarding functional foods through an integrative and thorough assessment of different involvement types and advertising claims as well as consumers’ residential regions. This enhanced involvement model with market segmentation considerations can be applied in studies on various consumption behaviours.

Based on these results, we offer five practical suggestions that governmental agencies might use to formulate appropriate management policies for protecting consumer rights and public health. First, because consumers pay more attention to functional foods, they more actively search for information and enter purchasing situations; therefore, government agencies must employ various channels in the provision of accurate and sufficient health-related information and proactively issue lists of products violating advertising regulations to enhance consumer awareness of specific products. Second, products with health claims are more likely to induce the effect of purchasing situations on consumers and even generate purchasing behaviour. Therefore, government agencies must publicly stress that a single ingredient or product cannot fulfil specific healthcare functions and that those healthy physiological functions can be achieved through a balanced diet and active lifestyle. Third, people who engage with functional food advertisements are more likely to be affected in the purchasing context. To reduce consumers’ erroneous selection of products under the influence of advertising, government agencies must strengthen their monitoring and investigation of false advertisements to protect the rights and interests of consumers. Fourth, the lack of regional medical resources causes consumers to be more susceptible to advertising and purchasing situations. Government agencies must prioritise the provision of medical resources in the nonnorthern regions of Taiwan (including the outlying islands). This could include the development of remote consultation and assessment to compensate for the lack of medical resources and reduce the public’s use of functional food as an alternative medical plan. Finally, government agencies must formulate and implement healthcare policies based on the market segments identified in this study. For example, northern residents tend to be affected by advertising messages with disease risk reduction claims; hence, government agencies can monitor and investigate advertisements with such efficacy claims in the northern region. Only then can limited administrative resources be more effectively applied in areas with greater potential harm.

This study has the following four limitations: (1) This research focused on the general consumer, but the collection of questionnaires solely through the Internet may have resulted in sampling error. (2) This study divided advertising claims into the three categories of nutrition, healthcare, and reduction in disease risk. In regard to people’s demand for functional foods, different functional claims (such as muscle gain or fat loss) can be further subdivided to reflect different product features, concern about the advertised message, and feelings towards the purchasing context. (3) The hypothesis that regional differences can be noted in the intention to purchase functional foods was based on differences in regional medical resources, but this study only divided Taiwan into northern and nonnorthern regions; thus, we were unable to detect detailed differences among various counties and cities. (4) The predictive validity of the mediating model was low; therefore, we could not fully explain the factors affecting the intention to purchase functional foods.

Based on the aforementioned limitations, future research can employ additional data-collecting methods such as mail-in surveys, in-person interviews, and telephone interviews. With these methods, the number and structure of valid questionnaires may more fully represent the Taiwanese population. Regarding the problem of the division of residential regions, the townships could be added in a revised questionnaire, and the regional division could be classified according to the availability of medical resources as reported by government agencies. In addition, different consumer groups address diverse needs through the purchase of functional foods; hence, health-related claims could be further divided to cover more consumer need categories. Finally, additional predictive variables (such as health conditions, consumer attitudes, purchase motivation, perceived risk, and subjective norms) must be taken into account to deepen our understanding of the influential factors and their effects on purchase intention. This research can assist in the construction of a thorough theoretical framework and the implementation of policies to prevent misleading advertisement claims.

## Figures and Tables

**Figure 1 foods-11-00978-f001:**
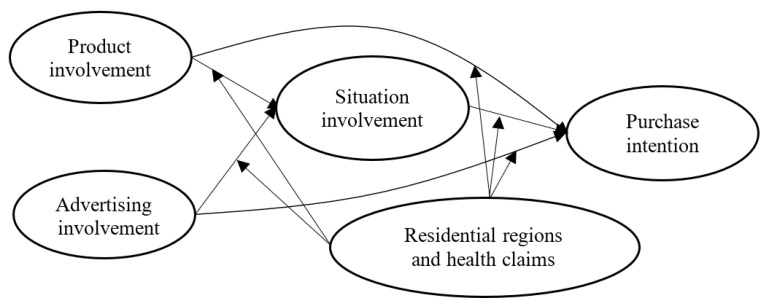
Conceptual model of the relationships between involvement types and purchase intention.

**Figure 2 foods-11-00978-f002:**
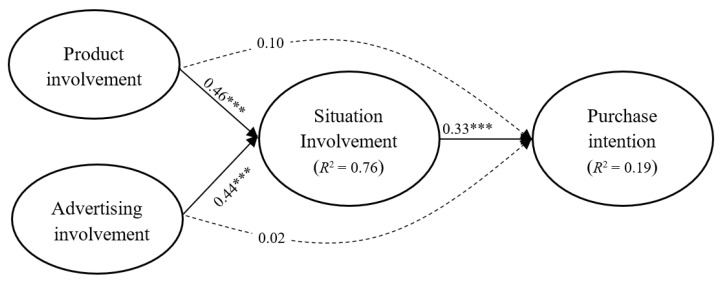
Mediation model for the relationship between involvement types and purchase intention (*n* = 1046) *** *p* < 0.001.

**Figure 3 foods-11-00978-f003:**
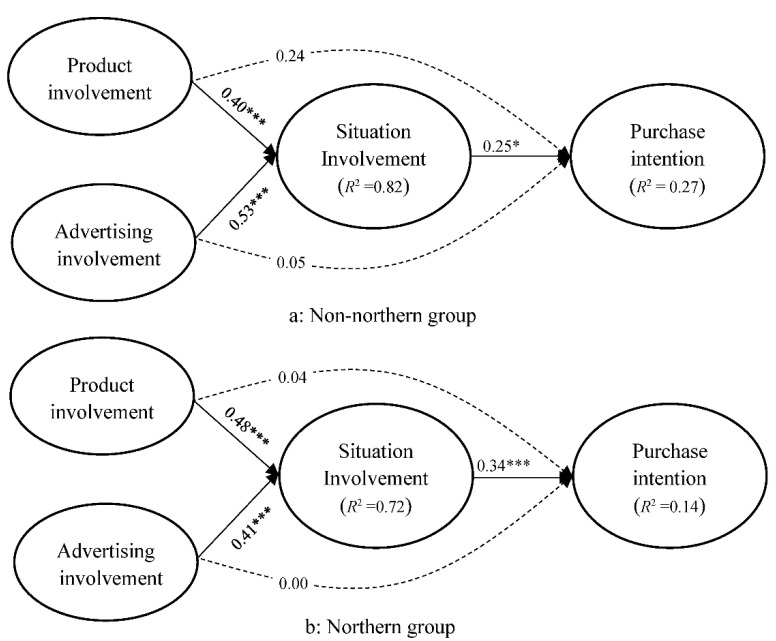
Moderation model for purchase intention in the nonnorthern region ((**a**), *n* = 662) and northern region groups ((**b**), *n* = 384) * *p* < 0.05, *** *p* < 0.001.

**Figure 4 foods-11-00978-f004:**
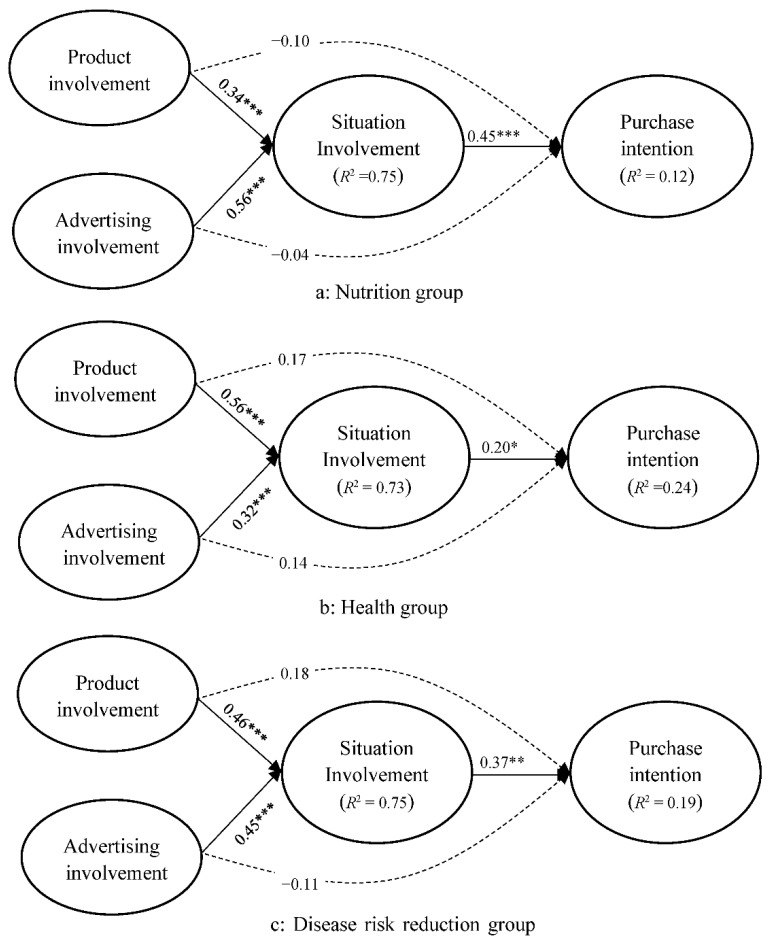
Moderation model for purchase intention (*n* = 1046) in the nutrition claim ((**a**), *n* = 350), health claim ((**b**), *n* = 350), and disease risk reduction claim groups ((**c**), *n* = 346) * *p* < 0.05, ** *p* < 0.01, *** *p* < 0.001.

**Table 1 foods-11-00978-t001:** Respondent demographics (*n* = 1046).

Demographic Variables	National Census (above 15 years/old)	Percentage (Frequency)		
Claims		Nutrition	Health	Disease risk reduction
	21,597,840	33.5% (350)	33.5% (350)	33.1% (346)
Gender				
Male	10,134,705	30.6% (107)	34.0% (119)	28.3% (98)
Female	10,463,135	69.4% (243)	66.0% (231)	71.7% (248)
Age				
15–35 years old	5,830,865	38.6% (135)	36.3% (127)	33.2% (115)
36–49 years old	5,671,100	38.3% (134)	36.7% (128)	44.3% (153)
50 years old and above	9,095,875	23.1% (81)	27.0% (95)	22.5% (78)
Education Levels				
Senior high school and lower	13,436,711	6.0% (21)	3.4% (12)	6.1% (21)
University or junior college	6,913,223	50.3% (176)	47.1% (165)	56.6% (196)
Graduate school and higher	1,247,906	43.7% (153)	49.5% (173)	37.3% (129)
Residential Regions				
Northern Regions	9,327,807	67.1% (235)	62.6% (219)	60.1% (208)
Nonnorthern Regions	12,270,033	32.9% (115)	37.4% (131)	39.9% (138)

Note: 2020 Taiwan census information was retrieved from: https://www.ris.gov.tw/app/en (accessed on 1 January 2022).

**Table 2 foods-11-00978-t002:** Confirmatory factor analysis (*n* = 1046).

Item/Variable	Product Involvement	Advertisement Involvement	Situational Involvement	Purchase Intention
Factor loadings				
Item 1	0.83	0.85	0.81	0.91
Item 2	0.82	0.86	0.85	0.97
Item 3	0.82	0.90	0.84	0.89
Item 4	0.88	0.83	0.89	0.81
Item 5	0.89	0.70	0.81	
Item 6	0.86	0.62	0.61	
Item 7	0.87			
Item 8	0.86			
α	0.96	0.91	0.91	0.94
Composite reliability	0.96	0.91	0.92	0.94
AVE	0.73	0.64	0.65	0.80

**Table 3 foods-11-00978-t003:** Heterotrait–monotrait ratio of correlation (*n* = 1046).

Item/Variable	Product Involvement	Advertisement Involvement	Situational Involvement	Purchase Intention
Product involvement	(0.85)			
Advertisement involvement	0.86	(0.80)		
Situational involvement	0.85	0.85	(0.81)	
Purchase intention	0.41	0.38	0.46	(0.90)

Note: Parenthetical values indicate the square root of the average variance extracted.

**Table 4 foods-11-00978-t004:** Comparative results of residential regions under the same advertising claim (*n* = 1046).

Test Results/Advertising Claims	Nutrition	Health	Disease Risk Reduction
Configural invariance	Appropriate fit (*χ*^2^/*df* = 2.24, RMSEA = 0.06, SRMR = 0.04, CFI = 0.92, TLI = 0.91)	Appropriate fit (*χ*^2^/*df* = 2.38, RMSEA = 0.07, SRMR = 0.06, CFI = 0.92, TLI = 0.91)	Appropriate fit (*χ*^2^/*df* = 2.15, RMSEA = 0.06, SRMR = 0.04, CFI = 0.93, TLI = 0.92)
Metric invariance	Insignificant difference (Δ*χ*^2^ = 18.70, Δ*df* = 20, *p* > 0.05)	Insignificant difference (Δ*χ*^2^ = 19.97, Δ*df* = 20, *p* > 0.05)	Insignificant difference (Δ*χ*^2^ = 28.00, Δ*df* = 20, *p* > 0.05)
Invariance results	Full invariance	Full invariance	Full invariance
Multigroup analysis	Significant change in *χ*^2^ (Δ*χ*^2^ = 6.68, Δ*df* = 1, *p* < 0.05)	Significant change in *χ*^2^ (Δ*χ*^2^ = 4.56, Δ*df* = 1, *p* < 0.05)	Significant change in *χ*^2^ (Δ*χ*^2^ = 0.33, Δ*df* = 1, *p* > 0.05)
Nonnorthern group (situational involvement → purchase intention)	*β* = 0.68, *p* < 0.001, f^2^ = 0.09	*β* = 0.21, *p* < 0.001, f^2^ = 0.02	*β* = 0.29, *p* < 0.001, f^2^ = 0.02
Northern group (situational involvement → purchase intention)	*β* = 0.34, *p* < 0.001, f^2^ = 0.03	*β* = 0.36, *p* < 0.001, f^2^ = 0.04	*β* = 0.47, *p* < 0.001, f^2^ = 0.07

## Data Availability

Data are available upon request.

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
