# Peer review of "Involvement Theory with Market Segmentation: Effect of False Functional Food Advertising on Purchase Intention"

_foods, 2022, doi:10.3390/foods11070978_

Round 1

Reviewer 1 Report

I think the topic addressed is very interesting and very relevant. In fact, what is pointed out in lines 56-58 is very important. I highly recommend the authors to work on this topic, since it is of great importance at the level of individuals and society, as well as for health.

Notwithstanding the above, I consider that there are some issues to be solved.

Minor issues:

Lines 16-17. There are not 1,046 "valid samples" but responses from a sample. Is it correct? Please, explain.

Line 19. The reason for dividing Taiwan into two regions (non-northern vs. northern) is expected to be explained.

Line 22. Two different issues: nutritional and health claims. At this moment, readers do not know whether these issues are linked or are independent one each other.

Line 50. The wording used is not correct. Some of these studies should be quoted, at least two as examples.

Line 67. The study is referring to Taiwan. However, readers will be from anywhere in the world. Some reference to other settings (The US, Europe, Japan, South America, and others) is of interest.

Line 127. The statement "Food consumption is typically a low-involvement activity" is very debatable. Your sentence is true for repetitive purchase products (vegetables, meat, fish), but it is not true for specific consumers' lifestyles. For an omnivore, the purchase of vegetables is routine, but it becomes high-involvement if we refer to vegetarians or vegans. There are high or low involvement foods. Read the following documents (DOIs):

10.1016/j.phanu.2019.100157

10.1108/00070709510085648

10.1002/mar.20302

10.1080/10454446.2019.1629558.

Line 218 (Figure 1). Why does the moderation refer only to one relationship and not to the whole model? A convincing explanation is missing, not only of the relationship, but also of why the rest of the relationships are not moderated with "purchase intention" variable.

Line 280. For non-Chinese readers who do not know Chinese, it would be important to include an appendix with the questionnaire used.

Major issues:

Lines 283-284. It is not explained how the fieldwork was conducted, how the participants were contacted, how the quality of the responses was controlled and how the responses received were debugged.

Line 307. Section 4.2. The authors introduce the quality of the model within the results. This is not appropriate. A section should be included to indicate the reliability and validity of the constructs, as well as the model fit. The results should then be presented in the next section.

Lines 308-313. The authors use SEM as if the sample meets the multivariate normality condition. However, I have not found Mardia's coefficient or the one used by AMOS to test for such multivariate normality. If this is not present, robust statistics should be presented.

In several places in the text, when presenting statistical results, in addition to stating the t-test probability you have to include the effect size.  The probability says little or nothing about the relevance of the findings (see http://www.math.chalmers.se/Stat/Grundutb/GU/MSA220/S18/effectsize.pdf).

I do not understand the point of including so much graphical modeling when it can be more parsimoniously resolved with a table.

Lines 712-717. There is a minor conceptual error. The conclusion section is not a place to emphasize literature analysis. I recommend that these references be removed and paraphrased, relating them to the authors' conclusions, which is what is important.

Reviewer 2 Report

The research deals with an interesting topic and is based on a solid sample of consumers. Though results might have some territorial limitations, they can expect international interest.

I have the following suggestions for the improvement of the manuscript:

  • Though the topic of functional foods receives great attention in the recent literature on food marketing, the literature review mainly refers to quite old references. More recent findings should be also included.
  • In Figure1 some of the text is cut and hard to read.
  • Though the hypotheses are well structured in the section of the literature review, the results part don't follow this logic and is hard to follow.
  • The high number of path analyses (Fig. 2-7) makes the paper difficult to follow, maybe the number of figures might be reduced.
  • Though it is also listed among the limitations, it is not really validated why the territorial distribution is divided into northern and non-northern parts.

Reviewer 3 Report

Thank you very much for the opportunity to review this manuscript.

The paper is well organized and well written.

Some minor remarks are follow.

Please write the bibliography as required by the journal instructions.Both in the text and in the list references.
